# A cohort study on the association between changing occupational stress, hair cortisol concentration, and hypertension

Jin Wang[1], Lejia Zhu[1], Lin Song[1], Ziqi Zhou[1], Weiling Chan[1], Geyang Li[1], Li Zhou[1], Jing Xiao[2], Yulong Lian[1] *

1 Division of Epidemiology and Medical Statistics, School of Public Health, Nantong University, Nantong, Jiangsu, China, 2 Department of Occupational Environmental Toxicology, School of Public Health, Nantong University, Nantong, Jiangsu, China

* lianyulong444@163.com

**Data Availability Statement:** There are ethical restrictions on publicly sharing the data for this study. The datasets used and/or analyzed during the current study are available from the

## Abstract

### Objective

To explore the relationship between changing occupational stress levels, hair cortisol concentration (HCC), and hypertension.

### Methods

Baseline blood pressure of 2520 workers was measured in 2015. The Occupational Stress Inventory—Revised Edition (OSI-R) was used to assess changes in occupational stress. Occupational stress and blood pressure were followed up annually from January 2016 to December 2017. The final cohort numbered 1784 workers. The mean age of the cohort was 37.77±7.53 years and the percentage male was 46.52%. At baseline, 423 eligible subjects were randomly selected for hair sample collection to determine cortisol levels.

### Results

Increased occupational stress was a risk factor for hypertension [risk ratio (RR) = 4.200, 95% confidence interval (CI): 1.734–10.172]. The HCC of workers with elevated occupational stress was higher than that of workers with constant occupational stress [(ORQ score ≥70: geometric mean±geometric standard deviation = 5.25±3.59 ng/g hair; 60–90: 5.02 ±4.00; 40–59: 3.45±3.41; <40: 2.73±3.40) $x^2$ = 5.261]. High HCC increased the risk of hypertension (RR = 5.270, 95% CI: 2.375–11.692) and high HCC was associated with higher rates of elevated diastolic and systolic blood pressure. The mediating effect of HCC was 0.51[(95% CI: 0.23–0.79, odds ratio(OR) = 1.67] and accounted for 36.83% of the total effect.

### Conclusions

Increased occupational stress could lead to an increase in hypertension incidence. High HCC could increase the risk of hypertension. HCC acts as a mediator between occupational stress and hypertension.

corresponding author via email
(lianyulong444@163.com) or the author's
institution via email (ggwsxy@ntu.edu.cn) upon
reasonable request.

**Funding:** This work was supported by the Natural
Science Foundation of Jiangsu Province, China
(Grant Number: BK20171256, BK2021020829);
Qinglan Project of Jiangsu Province of China. The
funders had no role in study design, data collection
and analysis, decision to publish, or preparation of
the manuscript. There was no additional external
funding received for this study.

**Competing interests:** The authors declare that
there are no conflicts of interest.

## 1. Introduction

Surveys in recent years have shown a significant upward trend in the global prevalence of hypertension, which is expected to affect about 30% of the global population by 2025 [1, 2]. In 2018, there were 245 million people with hypertension in China, and this number is still rising [3]. Reducing the prevalence of hypertension-related disease has become one of the most important public health challenges worldwide.

A quantitative meta-analysis conducted by Landsbergis et al. [4] found that when compared with people exposed to lower occupational stress, people with high occupational stress had a mean increase in systolic blood pressure of 3.43 mmHg [95% confidence interval (CI): 2.02–4.84] and a mean increase in diastolic blood pressure of 2.07 mmHg (95% CI: 1.17–2.97). In a cross-sectional study conducted by Shan-Fa et al. [5], high occupational stress as measured using the Effort-Reward Imbalance Questionnaire (ERI-Q) was associated with an increased prevalence of hypertension in male workers. A 1:1 matched case-control study conducted by Jian Li et al. [6]. in 212 pairs of university staff members measured occupational stress using the self-reported Job Content Questionnaire (JCQ). The prevalence of hypertension of those with medium occupational stress was twice as high (95% CI: 1.003–4.193) and of those with high occupational stress 2.87 times as high as those with low occupational stress (95% CI: 1.142–7.194).

Some occupations require repetitive tasks to be performed over long periods and/or require high levels of concentration for prolonged periods, either or both of which may lead to increased levels of occupational stress. These occupations can be considered high-pressured. At present, most studies focus on traditional high-pressure occupations such as laborers, medical staff, teachers, and police. The working environment of different occupations has its particularity, and the impact on the occupational population is also different. In the study of Zhou et al. [7], 254 white-collar were used as subjects, and it was found that the occupational role questionnaire (ORQ) score of a hypertensive group was higher than that of a non-hypertensive group, suggesting occupational stress is a risk factor for hypertension. Jingui Wu et al. [8] conducted a cross-sectional study of different occupational groups (including service workers, state-owned enterprises, private enterprises, researchers, teachers, community medical workers and bus drivers), and found that some occupational groups may be more prone to occupational stress-related hypertension, such as teachers, community health care workers, researchers, and bus drivers.

Cortisol is a natural glucocorticoid secreted by the zona fasciculata and zona reticulata cells in the adrenal cortex, regulated by the hypothalamic-pituitary-adrenal (HPA) axis. The HPA axis is a major component of the psychophysiological stress response, and previous studies have shown that occupational stress can lead to increased cortisol secretion [9, 10]. The mechanisms through which cortisol increases blood pressure are as follows: (1) cortisol can increase the activity of phenyl ethanolamine N methyl transferase, inhibit the activity of catecholamine ox methyltransferase, and increase the content of adrenaline in plasma; (2) cortisol affects the expression of adrenal α receptor and enhances the effect of catecholamines but, also through the central nervous system, reverses regulation of adrenocorticotropin-releasing hormone secretion; (3) cortisol can inhibit the synthesis of prostaglandin, bradykinin, 5-hydroxytryptamine and histamine, causing vasoconstriction; (4) cortisol promotes the reabsorption of renal tubules, increase blood volume, thereby increasing blood pressure [11, 12].

However, because of the circadian rhythm of human cortisol secretion, cortisol concentration in blood, urine, and saliva can only reflect individual stress in recent minutes or hours [13], and repeated sampling is required. Compared with cortisol in blood, urine, and saliva, cortisol in hair samples (hair cortisol concentration; HCC) is characterized by noninvasive

sampling, easy storage of samples [14], and accumulative concentration [15]. It is less influenced by circadian rhythm and sampling methods [13] and can better reflect long-term cortisol exposure levels in the body [13, 16]. Experimental studies have demonstrated an increase in blood pressure following an experimental infusion of cortisol in men with normal blood pressure [17, 18]. A study by Bautista et al. [19] suggested that elevated HCC is a risk factor for hypertension. In the study of Shanfa et al. [20], occupational stress was found to influence levels of cortisol in the blood, urine, and saliva. In the study by Kim et al. [21], HCC in pregnant women was found to be related to perceived stress, and Ibar et al. [22] found a direct correlation between HCC and perceived stress, as well as between HCC and the emotional exhaustion component of burnout. However, the relationship between occupational stress, high blood pressure, and cortisol levels as measured in the hair has not been studied, so controversy remains about the relationship between occupational stress, hypertension, and HCC.

Based on the cited studies, it is evident that research is required to explore the relationship between changing occupational stress and hypertension and whether HCC is a mediator. Therefore, in this cohort study, an occupational stress scale score, occupational health examination, and HCC in hair samples were used in an analysis of the relationship between high blood pressure, occupational stress, and hair cortisol. The following hypotheses were put forward: (1) increased occupational stress increases incidence of hypertension; (2) high HCC is associated with increased incidence of hypertension; (3) high HCC can lead to elevated blood pressure; and (4) HCC is a mediating variable between occupational stress and hypertension.

## 2. Materials and methods

### 2.1 Participants and procedures

In this study, all 2520 current employees of the Petroleum Administration Bureau and petrochemical companies in Xinjiang were selected as the research subjects and stratified according to the Classification Catalog of Petroleum and Petrochemical Jobs in China [23]. A questionnaire survey on occupational stress and a physical examination were conducted for selected on-the-job personnel at the Center for Disease Control and Prevention, Karamay City, Xinjiang Province in 2015. After excluding 403 patients with hypertension and coronary heart disease during the baseline period, the remaining 2116 subjects were followed up with an occupational stress questionnaire and occupational health examination once a year in 2016 and 2017, and final follow-up was conducted in December 2017. A total of 332 people were lost to follow-up during the follow-up period, a loss of 15.69%. Finally, a total of 1784 people that included 830 males and 954 females were included in the analysis of occupational health examination results and occupational stress scale measurements. In addition, 423 eligible subjects were randomly selected at baseline for hair sample collection and measurement of HCC. Characteristics identified for eligibility for selection were: (1) hair length greater than 3 cm; (2) non-hypertensive; (3) no perm, hair dye, or frequent shampooing.

### 2.1.1 Occupational stress

The Revised Occupational Stress Inventory (OSI-R) proposed and developed by Osipow et al. [24] was used to measure occupational stress in this study. The scale is divided into three subscales: ORQ, personal strain questionnaire (PSQ), and personal resources questionnaire (PRQ), with a total of 140 items. Among the subscales, the ORQ reflects the level of occupational stress experienced by individual workers and includes multiple subitems such as role overload (RO), role insufficiency (RI), role ambiguity (RA), role boundary (RB), responsibility (R), and physical environment (PE). Each subitem is composed of 10 subitems, and each subitem is scored from 1 to 5. The ORQ scale was most meaningful to this study, so we chose to use it.

The original score of each subitem of occupational stress was transformed into a total score T score with a mean of 50 and a standard deviation of 10 (T = 50+10×(x-x)/s) [25], where x was the original score and s was the standard deviation), and this was used to evaluate the degree of occupational stress. The T value of each subitem of the occupational stress norm of technical workers was used to grade the degree of occupational stress, and the occupational stress of the research object was evaluated [25]. The study was divided into four corresponding grades according to the T score. The occupational stress classification was as follows: high occupational stress (T≥70), medium occupational stress (60≤T≤69), normal occupational stress (59≥T≥40), and lack of occupational stress (T<40).

At follow-up, there were three possible changes in T score: increased, decreased, and unchanged. Increased status was divided into (baseline-follow-up) lack-normal, lack-medium, lack-high, normal-medium, normal-high, and medium-high. Decreased status was divided into high-medium, high-normal, high-lack, medium-normal, medium-lack, and normal-lack. Stable (unchanged) status was high-high, medium-medium, normal-normal, or lack-lack. These represented increase, decrease, and stable ORQ score [25].

**2.1.2 Hair cortisol concentration.** Hair samples of length 2–3 cm and weight 20–30 mg were collected from the hair roots of the participants. The pretreatment process for hair sampling followed the experimental procedures described in the patent "A pre-treatment method for detecting cortisol content in hair" [26]. The hair samples were washed and exfoliated with 2–3 mL isopropanol for 5 minutes; the exfoliated hair samples were frozen with liquid nitrogen for more than 4 hours and then crushed; and the crushed hair samples were placed in centrifuge tubes with 5 mL methanol solution and 3 mL ether solution (methanol: ether volume ratio 5:3) added and placed in a water bath at 50.8˚C for 16 h for extraction and incubation. The analytical process involved mixing the hair debris by inversion several times and centrifugation at low speed (3500 rpm) for 15 minutes. The super-natant was removed and put it into a 4 mL Eppendorf (EP) tube; a nitrogen blower was used to dry the mixture after extraction; 2 mL phosphate buffer solution was added, and the sample placed in a refrigerator at -4˚C for cold storage until the test day. A radioimmunoassay kit for detecting iodine [125I] cortisol and an automatic radioimmunoassay instrument were used to determine HCC.

**2.1.3 Hypertension.** After the subject had rested for 30 minutes, their blood pressure was measured three times consecutively while seated, with the readings taken at least 5 minutes apart. The average of the second and third measurement was recorded. Subjects with systolic blood pressure ≥140 mmHg and (or) systolic blood pressure ≥90 mmHg were diagnosed as hypertensive according to the Chinese Guidelines for the Prevention and Treatment of Hypertension [27].

**2.1.4 Covariates.** The covariates in this study included general demographic variables (age, sex, history of hypertension, coronary heart disease, stroke, diabetes, and family history of cancer), physical exercise level, alcohol consumption, smoking, and body mass index. According to the frequency of the subjects' weekly participation in physical exercise, they were divided into three categories: no exercise, exercise <3 times/week, exercise ≥3 times/week, and irregular weekly exercise. Frequency of alcohol consumption was divided into three groups: nondrinking, drinking, and quit drinking. Frequency of smoking was divided into the categories of nonsmoking, smoking, and quit smoking. The height and weight of the subjects were measured, and their body mass index (BMI) (kg/m$^2$) was calculated and categorized as normal (<25) or overweight (≥25).

## 2.2 Ethical considerations

All the subjects provided signed informed consent after receiving research-related information. The study was approved by the ethics committee of Nantong University.

### 2.3 Statistical analysis

Epidata 3.0 was used to establish a database for double-key entry of the questionnaire data, and SPSS 20.0 was used to collate and analyze the data. Measurement data were described by $\bar{X} \pm S$ and interquartile range [M (Q1–Q3)], and geometric mean concentration (GM)±GSD was used to improve statistical power. The $x2$ test was used to compare the enumeration data, and t-test or analysis of variance was used to compare the measurement data. Two logistic regression models were devised using the enter method. No adjustments were made in Model 1, whereas Model 2 adjusted for age, sex, smoking, alcohol consumption, BMI, physical exercise, family history (hypertension, diabetes, coronary heart disease, stroke, tumor history), and other factors. The significance level was α = 0.05.

We conducted a mediation effect analysis to understand the mechanism by which one variable affected the other. The correlation coefficient between occupational stress and hypertension indicated the overall effect. When HCC was a mediating variable, the effect coefficient of occupational stress on hypertension was a direct effect. The mediating effect was calculated by subtracting the direct effect from the total effect [28]. The significance of the HCC effect was verified using the method described by Karlson, Holm, and Brin [29]. If both the overall and indirect effects were significant, but the direct effects were not significant, HCC was considered to modulate the relationship between occupational stress and hypertension [30, 31]. We used this method to estimate the percentage of direct effects mediated by HCC.

## 3. Results

**Table 1** shows that the cumulative incidence of hypertension was 23.5% in males and 10.9% in females, and the difference was statistically significant (P = 0.000). Differences in each exposure group in age, BMI, smoking, drinking, family history of hypertension, coronary heart disease, stroke, and diabetes were statistically significant. The HCC of males was higher than that of females and the difference was statistically significant (P<0.05). Except for smoking and BMI, there were no statistical differences in HCC among different demographic characteristics.

**Table 2** shows that when the level of occupational stress increased, the cumulative incidence of hypertension was higher (P<0.05). Both Model 1 and Model 2 showed that, in contrast to an unchanged occupational stress (ORQ) score, a rise in ORQ was a risk factor for hypertension [risk ratio (RR) = 4.200, 95% CI: 1.734–10.172]. The higher the concentration of HCC, the higher the cumulative incidence of hypertension, with specific statistical significance (P<0.05). Both Model 1 and Model 2 showed that compared with intermediate HCC [first quartile (Q1)–Q2], higher HCC [Q2–Q3 (adjusted RR = 5.424, 95%CI: 1.016–28.966) and ≥Q3 (4.288 ng/g hair; adjusted RR = 39.080, 95%CI: 8.123–188.014)] and lower HCC [<Q1 (1.7347 ng/g hair; adjusted RR = 6.246, 95%CI: 1.160–33.627)] were both risk factors for hypertension.

As can be seen from **Table 3**, the changes in systolic and diastolic blood pressure correlated with the high and low levels of HCC, and the difference was statistically significant (P<0.05).

**Table 4** shows that with an increase in occupational stress (ORQ) score, HCC increased gradually, and the difference was statistically significant (P<0.05).

**Table 5** shows that occupational stress was associated with hypertension (B = 1.39, 95%CI: 0.63–2.14, OR = 4.01, P<0.001), and occupational stress was associated with HCC (B = 1.517, 95%CI: 1.352–1.681, P<0.05) and hypertension (B = 0.105, 95%CI: 0.075–0.134, P<0.05). When HCC was added as a mediator, the regression coefficients remained significant (B = 0.88; 95%CI: 0.12–1.63; OR = 2.40, P<0.05), and the mediating effect of HCC was 0.51 (95%CI: 0.23–0.79, OR = 1.67, P<0.001). HCC played a partial mediating role between

**Table 1. Comparison of hypertension and hair cortisol concentration by demographic characteristic.**

| variables | hypertension | | P | n(%) | HCC (ng/g) | P |
|---|---|---|---|---|---|---|
| | no (n(%)) | yes (n(%)) | | | ($\bar{X} \pm$ S) | |
| age | | | 0.001 | | | 0.478 |
| <30 | 290(89.2) | 35(10.8) | | 76(18.1) | 3.9406±3.9400 | |
| 30–40 | 516(84.5) | 95(15.5) | | 151(36.0) | 3.3838±2.7863 | |
| >40 | 679(80.1) | 169(19.9) | | 192(45.8) | 3.6874±3.6038 | |
| sex | | | 0.000 | | | 0.000 |
| male | 635(76.5) | 195(23.5) | | 158(37.7) | 4.8640±4.8176 | |
| female | 850(89.1) | 104(10.9) | | 261(62.3) | 2.8732±1.7460 | |
| Drinking | | | 0.003 | | | 0.158 |
| nondrinking | 774(86.0) | 126(14.0) | | 213(50.8) | 3.3261±2.6290 | |
| drinking | 655(81.0) | 154(19.0) | | 186(44.4) | 3.9783±4.1570 | |
| quit drinking | 56(74.7) | 19(25.3) | | 20(4.8) | 3.4995±2.5323 | |
| Smoking | | | 0.000 | | | 0.015 |
| nonsmoking | 951(87.1) | 141(12.9) | | 274(65.4) | 3.2882±2.5159 | |
| smoking | 468(77.9) | 133(22.1) | | 123(29.4) | 4.1608±4.5847 | |
| quit smoking | 66(72.5) | 25(27.5) | | 22(5.3) | 4.8028±4.5630 | |
| BMI | | | 0.000 | | | 0.000 |
| <25 | 1010(91.0) | 100(9.0) | | 279(66.6) | 3.1832±2.8701 | |
| ≥25 | 475(70.5) | 199(29.5) | | 140(33.4) | 4.5023±4.1330 | |
| Exercise | | | 0.059 | | | 0.659 |
| 0 | 358(86.1) | 58(13.9) | | 97(23.2) | 3.4903±3.1448 | |
| <3 | 526(82.7) | 110(17.3) | | 130(31.0) | 3.4438±3.2521 | |
| ≥3 | 176(86.7) | 27(13.3) | | 45(10.7) | 4.1255±3.7590 | |
| disorder | 425(80.3) | 104(19.7) | | 147(35.1) | 3.7178±3.5812 | |
| Family history of hypertension | | | 0.002 | | | 0.325 |
| yes | 428(81.5) | 97(18.5) | | 115(27.4) | 3.5182±3.3451 | |
| no | 970(83.0) | 199(17.0) | | 290(69.2) | 3.7260±3.4828 | |
| Not clear | 87(96.7) | 120(3.3) | | 14(3.3) | 2.3786±1.2205 | |
| Family history of coronary heart disease | | | 0.015 | | | 0.068 |
| yes | 48(73.8) | 17(26.2) | | 14(3.3) | 5.6181±6.9100 | |
| no | 1326(83.1) | 270(16.9) | | 380(90.7) | 3.5813±3.2871 | |
| Not clear | 111(90.2) | 12(9.8) | | 25(6.0) | 3.1549±1.4253 | |
| Family history of stroke | | | 0.004 | | | 0.544 |
| yes | 40(93.0) | 3(7.0) | | 7(1.7) | 2.9525±1.7878 | |
| no | 1364(82.4) | 291(17.6) | | 399(95.2) | 3.6647±3.4638 | |
| Not clear | 81(94.2) | 5(5.8) | | 13(3.1) | 2.7336±1.2262 | |
| Family history of diabetes | | | 0.015 | | | 0.304 |
| yes | 187(87.4) | 27(12.6) | | 42(10.0) | 2.9672±2.2337 | |
| no | 1226(82.2) | 266(17.8) | | 363(86.6) | 3.7247±3.5474 | |
| Not clear | 72(92.3) | 6(7.7) | | 14(3.3) | 2.9814±1.7288 | |
| Tumor history | | | 0.055 | | | 0.245 |
| yes | 70(88.6) | 9(11.4) | | 17(4.1) | 2.3478±1.5890 | |
| no | 1342(82.6) | 283(17.4) | | 389(92.8) | 3.6953±3.4881 | |
| Not clear | 73(91.3) | 7(8.8) | | 13(3.1) | 3.1562±1.6558 | |

**Table 2. Relationship between hypertension and occupational stress and HCC demonstrated by logistic regression analysis.**

| | Hypertension | Model 1 | | Model 2 | |
|---|---|---|---|---|---|
| | incidence/total(%) | RR (95% CI) | P | RR* (95% CI) | P |
| **Occupational stress** | | | | | |
| Unchanged | 6/79(7.6) | 1.000 | — | 1.000 | — |
| Decrease | 81/806(10.0) | 1.359(0.573–3.223) | 0.486 | 1.494(0.607–3.678) | 0.382 |
| Increase | 212/899(23.6) | 3.754(1.610–8.754) | 0.002 | 4.200(1.734–10.172) | 0.001 |
| **HCC (ng/g)** | | | | | |
| <1.7347 | 9/104(8.7) | 4.879(1.028–23.156) | 0.046 | 6.246(1.160–33.627) | 0.033 |
| 1.7347- | 2/105(1.9) | 1.000 | — | 1.000 | — |
| 2.8833- | 10/105(9.6) | 5.421(1.158–25.377) | 0.032 | 5.424(1.016–28.966) | 0.048 |
| 4.2888- | 43/105(41.0) | 35.718(8.359–152.627) | 0.000 | 39.080(8.123–188.014) | 0.000 |

Model 1: Single factor analysis.

Model 2: Adjusted for age, sex, smoking, drinking, body mass index, exercise, family history of hypertension, family history of diabetes, stroke, family history of coronary heart disease, and history of tumor.

CI: Confidence interval; RR: risk ratio.

occupational stress and hypertension. The mediating effect was significant, accounting for 36.83% of the difference between groups.

## 4. Discussion

This study found that increased level of occupational stress was a risk factor for hypertension, and that increased levels of occupational stress lead to high HCC. We also found that workers with high HCC had a higher incidence of hypertension than those with normal HCC. High HCCs were associated with higher levels of systolic and diastolic blood pressure. HCC acted as a partial mediator in the relationship between occupational stress and hypertension.

Several epidemiological studies show that long-term chronic occupational stress, which is a risk factor for the occurrence of hypertension, may lead to increased blood pressure [32–34]. A study from Japan [35] reported that work stress increased the risk of hypertension threefold. Liu et al. [36] also reported that teachers' occupational stress was associated with hypertension. Zhou et al. [7] found in their study of white-collar that these workers had high occupational stress, which is the main risk factor for hypertension. The study used the same occupational stress scale as our study, and these findings are consistent with our findings based on a population of workers. In addition, studies have established the mechanism of the increase in the incidence of hypertension caused by long-term occupational stress: under long-term occupational stress, the abnormal excitation of the sympathetic nervous system causes excessive secretion of epinephrine, norepinephrine, and other hormones through the sympatho-adrenal medulla

**Table 3. The relationship between systolic and diastolic pressure and hair cortisol concentrations.**

| | systolic pressure | | | $X^2$ | P | diastolic pressure | | | $X^2$ | P |
|---|---|---|---|---|---|---|---|---|---|---|
| | decrease | invariant | increase | | | decrease | invariant | increase | | |
| HCC (ng/g) | | | | 19.731 | 0.003 | | | | 23.253 | 0.001 |
| <1.7347 | 40(38.5) | 3(2.9) | 61(58.7) | | | 43(41.3) | 6(5.8) | 55(52.9) | | |
| 1.7347- | 29(27.6) | 0(0.0) | 76(72.4) | | | 42(40.0) | 4(3.8) | 59(56.2) | | |
| 2.8833- | 19(18.3) | 4(3.8) | 84(77.9) | | | 32(30.8) | 5(4.8) | 67(64.4) | | |
| 4.2888- | 18(17.1) | 3(2.9) | 84(80.0) | | | 15(14.3) | 6(5.7) | 84(80.0) | | |

**Table 4. Comparison of hair cortisol concentrations by ORQ score.**

| variables | n | HCC (ng/g) | F | P |
|---|---|---|---|---|
|  |  | ($\bar{X} \pm S$) |  |  |
| **occupational stress (ORQ score)** |  |  | 5.261 | 0.001 |
| ≥70 | 12 | 5.2477±3.5903 |  |  |
| 60–69 | 52 | 5.0192±3.9990 |  |  |
| 40–59 | 311 | 3.4542±3.4105 |  |  |
| <40 | 44 | 2.7320±3.3980 |  |  |

axis, resulting in increased myocardial contractility, increased heart rate, peripheral vascular contraction, and elevated blood pressure [37, 38]. Thus, we preliminarily speculate that with an increase in occupational stress, systolic and diastolic blood pressure and thus prevalence of hypertension also increase.

In our study, occupational stress was significantly correlated with higher HCC. One study found a significant positive relationship between HCC and self-reported stress levels in healthy pregnant women [39]. In 2014, Steinisch et al. [40] conducted a novel cross-sectional study on the relationship between occupational stress and HCC. They showed that HCC was significantly correlated with occupational stress. In addition, studies have found that the secretion of cortisol is affected by social pressure [41], and psychological stress can cause the activation of the hypothalamic-pituitary-adrenal (HPA) axis, resulting in increased cortisol synthesis [42]. These reports are consistent with our experimental results and have shown that measurement of HCC is a powerful tool for measuring chronic stress and can be used as an effective marker of long-term chronic stress [43, 44].

As the central control system of the human body, the HPA axis regulates the physiological functions of several important parts of the body. Cortisol can directly affect the central nervous system and the brain regions involved in the control of blood pressure (hypothalamus, limbic system, etc.). In addition to the brain, glucocorticoid receptors are present in the heart and vascular smooth muscle of resistance vessels, as well as in the kidneys, and thus directly affect blood pressure [45]. We found that workers with high HCC had higher prevalence of hypertension compared with workers with low and intermediate HCC. Hamer et al. [46] found that under conditions of occupational stress, the incidence of hypertension was correlated with cortisol level. These studies all demonstrated that exposure to high levels of cortisol may lead to increased blood pressure and the onset of hypertension, and this is consistent with the relationship between HCC and hypertension found in our study.

Few studies have explored the relationship between HCC and hypertension; however, existing studies have found that HCC was highly consistent with cortisol concentrations in saliva, urine, and blood over a long period of time [47–49]. This indirectly suggests that high HCC is a risk factor for hypertension over time.

**Table 5. The mediating effect of hair cortisol concentration between occupational stress and hypertension.**

|  | hypertension | B (95%CI) | SE(β) | z | OR (95%CI) | p | (%) |
|---|---|---|---|---|---|---|---|
| **occupational stress** | **Total effect** | 1.39(0.63–2.14) | 0.38 | 3.61 | 4.01(1.88–8.51) | <0.001 | 36.83 |
|  | **Direct effect** | 0.88(0.12–1.63) | 0.39 | 2.27 | 2.40(1.13–5.11) | 0.023 |  |
|  | **Indirect effect** | 0.51(0.23–0.79) | 0.14 | 3.59 | 1.67(1.26–2.20) | <0.001 |  |

%: Mediating effect as percentage of total effect.

CI: Confidence interval; OR: odds ratio; SE: standard error.

In this study, changing occupational stress, HCC, and hypertension were analyzed together for the first time. We studied changes to occupational stress separately and found respective associations with hypertension and HCC. We further explored this relationship and found that HCC was actually involved in mediating the relationship between occupational stress and hypertension. The mediating effect of HCC accounted for 36.83% of the relationship. We also found a mediating effect from HCC between occupational stress and hypertension that may effectively avoid the problems with blood, urine, and saliva cortisol detection methods and may better reflect long-term cortisol exposure in the body.

To the best of our knowledge, this is the first cohort study examining changes to occupational stress, hypertension, and HCC that further confirms the results of previous studies, with strong associations. However, some limitations were inevitable. First, although most confounders were measured and adjusted for in this study, confounders such as fat, sugar, and dietary fiber intake, and noise and high temperature in the work environment, were not measured. Second, HCC measurement was only performed at baseline and not at follow-up, and the number of participants was small, resulting in a small sample size that may have affected the results. Third, the Classification Catalogue of Petroleum and Petrochemical Jobs in China was used for stratified cluster sampling by job type and selection bias was unavoidable. Fourth, during the COVID-19 epidemic period, occupational stress may have changed due to the work environment or other factors, and further follow-up of etiological relationship between occupational stress and hypertension is required.

## 5. Conclusion

In conclusion, the study found that an increased level of occupational stress was a risk factor for hypertension, and high HCC was associated with increased levels of occupational stress. High HCCs were positively associated with increased blood pressure and the risk of hypertension. HCC acted as a partial mediator in the relationship between occupational stress and hypertension. Therefore, occupational stress is associated with increased blood pressure and increased risk of hypertension due to its influence on cortisol levels.

## Acknowledgments

We are appreciative of the help provided by Xinjiang Karamay CDC and Karamay Hospital's occupational diseases department.

## Author Contributions

**Conceptualization:** Yulong Lian.

**Data curation:** Lejia Zhu, Ziqi Zhou, Weiling Chan.

**Formal analysis:** Jin Wang, Lin Song, Geyang Li, Li Zhou, Yulong Lian.

**Funding acquisition:** Yulong Lian.

**Methodology:** Ziqi Zhou, Jing Xiao, Yulong Lian.

**Resources:** Jing Xiao, Yulong Lian.

**Software:** Jin Wang.

**Supervision:** Jing Xiao.

**Validation:** Lin Song.

**Writing – original draft:** Jin Wang.

**Writing – review & editing:** Yulong Lian.

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
