## [Decision Letter · Decision Letter 0]

5 Jan 2023

PONE-D-22-19724A cohort study on the association between changing occupational stress, hair cortisol concentration, and hypertensionPLOS ONE

Dear Dr. Lian,

Thank you for submitting your manuscript to PLOS ONE. After careful consideration, we feel that it has merit but does not fully meet PLOS ONE’s publication criteria as it currently stands. Therefore, we invite you to submit a revised version of the manuscript that addresses the points raised during the review process. I agree with both reviewers that this manuscript has potential value. However, as outlined by Reviewer 1, there are several major concerns related to bias in the (statistical) methodology and/or description of their input and results which make the manuscript difficult to interpret to its full extent. A future revision requires a very clear, thorough and exhaustive response through changes or rebuttal to the reviewer comments in order to consider the manuscript for further evaluation. Please also thoroughly revise the full manuscript for language, as suggested per the (non-exhaustive) list of suggestions provided by the reviewers.

We look forward to receiving your revised manuscript.

Kind regards,

Ralph C. A. Rippe, Ph.D.

Academic Editor

PLOS ONE

Journal Requirements:

“This work was supported by the Natural Science Foundation of Jiangsu Province, China (Grant Number: BK20171256、BK2021020829); Qinglan Project of Jiangsu Province of China.”

Reviewers' comments:

Reviewer's Responses to Questions

**Comments to the Author**

1. Is the manuscript technically sound, and do the data support the conclusions?

Reviewer #1: Partly

Reviewer #2: Yes

2. Has the statistical analysis been performed appropriately and rigorously? 

Reviewer #1: No

Reviewer #2: Yes

3. Have the authors made all data underlying the findings in their manuscript fully available?

Reviewer #1: No

Reviewer #2: Yes

4. Is the manuscript presented in an intelligible fashion and written in standard English?

Reviewer #1: No

Reviewer #2: Yes

5. Review Comments to the Author

Reviewer #1: This is an interesting study that benefits from a large sample size and longitudinal follow-up. However, have some concerns regarding the statistical analyses. Moreover, the readibility of the manuscript could be improved. I have the following suggestions for the authors:

1) My main concern is that by categorising the primary exposure and outcome variables (an approach that is not well explained), the authors are losing valuable information. This is partiicularly prolematic for the primary predictor, why not look at scores at baseline follow-up and their interaction rather than increase/decrease/stable. Moreover, categorisation poses problems for some variables due to small cell sizes - for example - in Table 4 the number of cases who developed hypertension are 9, 2, and 10 in the lower HCC groups. The categorisation of is particularly unclear for blood pressure, is a change in absolute levels classed as a decrease/increase or was a threshold for amount of change required? So few people classed as invariant implies that this might difference in absolute levels which would be highly inappropriate. Further details required.

2) Statistical analaysis section is unclear, terms are used that are not described and it is not clear how these relate to the variables examined. Stepwise regression is mentioned in the final sentence, not clear whether this is forward/backward or how threshold for inclusion determined. Stepwise models are problematic as they often yield inconsistent results depending on the decisions above, instead, deciding a priori whether confounders should be included is more appropriate.

Minor points

3) Abstract: please add cohort details in methods or results (e.g., mean age, percentage male). Methods slightly unclear.

4) Methods: the description of the occupational stress questionnaire could be clearer. Why was this subscale selected specifically are the other scales not relevant?

5) Methods: Was the hair sample collected from the root (ie plucked) or was it cut as close to the root as possible?

6) Results: the number of results tables is overwhelming, can those with the same outcome or predictor variable be combined?

7) Results: Chi squared test adds nothing but the descriptives show that the numbers with outcome are small for reference category

8) Table 7: if p indicates P value then values of 0 should be replaced with p<0.001

9) Discussion: Readability of the first paragraph of the discussion could be improved - sentences could flow better.

10) Discussion, line 235 - do the authors mean 'mental health workers' instead of 'mental workers'?

11) Overall the discussion could be more succinct but should also acknowledge studies which show contrasting findings to those observed in the current study.

12) Discussion includes nothing on the proposed mechanisms

13) There are typographical errors in the text, in particular whole words that have been hyphenated e.g., line 27 'dis-ease'

Reviewer #2: This manuscript was a very interesting read to me as I have been working on work stress and cortisol research over the last two years. Correlating hair cortisol concentration with hypertension and work stress was a great idea for a scientific manuscript. The statistical analyses appear accurate and sound and the manuscript as a whole is nearly publishable.

The list below includes any concerns I had with the manuscript as it currently stands. Many of them are grammatical in nature. Please be sure to not write in first person (using "we") in scientific literature and don't confuse correlation with causation.

Reviewer Notes

Line 27: No reason to have the hyphen in disease.

Line 29: Remove the period after “[4]”

Line 43: Please be more specific than saying “workers” work in traditionally high-pressure occupations.

Line 45: Unsure what you mean by “mental workers”.

Line 52: No need to have a hyphen in researchers

Line 70: Change “Compared with in blood…” as you have two prepositions back-to-back. I’d suggest “Compared with cortisol in blood…”

Line 88: Don’t write in first person.

Line 90: Add a comma after “study”

Great sample size

Line 112: No reason for hyphen in subjects

Line 155: Can you better define “irregular” physical exercise?

Line 156: Can you distinguish “no alcohol consumption” vs “abstinence”?

Line 157: Can you describe “smoking cessation”?

Line 223 & Line 224: change “level” to “levels”

Line 225: Don’t write in 1st person

Line 232: Move the short paragraph beginning in line 230 into the next paragraph that begins on line 235. Additionally, the two sentences in lines 230-234 are pretty redundant in purpose. I’d suggest editing these two sentences in some way.

Line 235: What are “mental workers”?

Line 239: Don’t end a sentence in a helping verb “was”.

Line 256: No need for the hyphen in the word service.

Line 278: No need for the hyphen in the word under.

Line 305: No need for the hyphen in the word required.

Line 308: Don’t write in 1st person.

Line 309 and Line 312: You can’t necessarily say occupational stress caused high HCC (Line 309) or that occupational tension caused high blood pressure (Line 312). You can simply state that they were correlated with one another.

Line 312: No need for the hyphen in the word increase.

6. PLOS authors have the option to publish the peer review history of their article (what does this mean?). If published, this will include your full peer review and any attached files.

Reviewer #1: No

Reviewer #2: **Yes: **Thomas Gerding, MPH ST/ASHM

---

## [Author Response · Author response to Decision Letter 0]

22 Mar 2023

Response to Reviewers

Thank you for your very valuable comments and useful suggestions. In response to your suggestions, we do the following point-to-point responses. With your comments and help, we believe that the quality of our articles will be greatly improved.

Reviewer #1: 

1) My main concern is that by categorising the primary exposure and outcome variables (an approach that is not well explained), the authors are losing valuable information. This is particularly prolematic for the primary predictor, why not look at scores at baseline follow-up and their interaction rather than increase/decrease/stable. Moreover, categorisation poses problems for some variables due to small cell sizes - for example - in Table 4 the number of cases who developed hypertension are 9, 2, and 10 in the lower HCC groups. The categorisation of is particularly unclear for blood pressure, is a change in absolute levels classed as a decrease/increase or was a threshold for amount of change required? So few people classed as invariant implies that this might difference in absolute levels which would be highly inappropriate. Further details required.

Response 1)：We were very sorry for the confusion and have elaborated the basis for division in the revised manuscript. (Page 9, Line 129) According to Yang XW’s [1] method of developing occupational stress norms and conversion tables for managers, ORQ scores were transformed into t scores to represent the degree of occupational stress, and t scores were graded. We have explained in the revised manuscript. (Page 9, Line 139-145) The rules for identifying changes in occupational stress during follow-up are shown in the table below.

 follow-up occupational stress

baseline occupational stress High middle normal lack of

High - ↓ ↓ ↓

middle ↑ - ↓ ↓

normal ↑ ↑ - ↓

lack of ↑ ↑ ↑ -

↑：increase 

↓：decrease 

-：stable

As the sample size of workers who collected hair in this study did not reach the minimum theoretical sample size, this limitation has been mentioned in the Discussion section of the revised manuscript. (Page 23, Line 310-312)

As in the study of Stalder T et al.[2], even if the value of blood pressure did not increase to the threshold, Elevated systolic and diastolic blood pressure were also associated with HCC. 

1. Yang XW, Wang ZM, Jin TY, Lan YJ. [Study of the occupational stress norm and it's application for the executive group and administrative support group]. Wei sheng yan jiu = Journal of hygiene research. 2006;35(4):477-80. Epub 2006/09/22. PubMed PMID: 16986528.

2. Stalder T, Steudte-Schmiedgen S, Alexander N, Klucken T, Vater A, Wichmann S, et al. Stress-related and basic determinants of hair cortisol in humans: A meta-analysis. Psychoneuroendocrinology. 2017;77:261-74. Epub 2017/01/31. doi: 10.1016/j.psyneuen.2016.12.017. PubMed PMID: 28135674.

2) Statistical analysis section is unclear, terms are used that are not described and it is not clear how these relate to the variables examined. Stepwise regression is mentioned in the final sentence, not clear whether this is forward/backward or how threshold for inclusion determined. Stepwise models are problematic as they often yield inconsistent results depending on the decisions above, instead, deciding a priori whether confounders should be included is more appropriate.

Response 2)： We felt sorry for the confusion. We used enter methods to build two logistic regression models and conduct mediation effect analysis. We have revised and supplemented it in the methods of the section revised manuscript. (Page 12, Line 195-204)

3) Abstract: please add cohort details in methods or results (e.g., mean age, percentage male). Methods slightly unclear.

Response 3): We would like to thank the reviewer for the comment and we have added it in Methods section of the manuscript. (Page 2, Line 8-9)

4) Methods: the description of the occupational stress questionnaire could be clearer. Why was this subscale selected specifically are the other scales not relevant?

Response 4): We felt sorry for the confusion, the Revised Occupational Stress Inventory (OSI-R) scale included three subscales: Occupational Role Questionnaire (ORQ), Personal Strain Questionnaire (PSQ), and Personal Resources Questionnaire (PRQ). The ORQ reflects the level of occupational stress experienced by individual workers, the PSQ reflects the individual stress response, and the PRQ reflects the ability to cope with stress response, so we chose to use the ORQ scale. [3].

3. J L, YJ L, ZM W, MZ W, MZ W, GQ L. The test of occupational stress inventory revised edition. Chinese Journal of Industrial Hygiene and Occupational Diseases. 2001;03:34-7.

5) Methods: Was the hair sample collected from the root (ie plucked) or was it cut as close to the root as possible?

Response 5): We collected 20-30mg hair samples from the 2 ~ 3cm of the root of the subject's hair.

6) Results: the number of results tables is overwhelming, can those with the same outcome or predictor variable be combined?

Response 6): We would like to thank the reviewer of this suggestion and we have merged the original Table 1 and Table 3 into the current table 1, and the original table 2 and Table 4 into the current table 2 in the revised manuscript.

7) Results: Chi squared test adds nothing but the descriptives show that the numbers with outcome are small for reference category

Response 7): Thank you for your question. We have modified the tables of the chi-square tests in the results (Tables 1 and 2). 

8) Table 7: if p indicates P value then values of 0 should be replaced with p<0.001

Response 8): We felt grateful for the comment and we have revised it in Table 7.

9) Discussion: Readability of the first paragraph of the discussion could be improved - sentences could flow better.

Response 9): Thanks for the informative comment. We have modified this paragraph in the discussion.

10) Discussion, line 235 - do the authors mean 'mental health workers' instead of 'mental workers'?

Response 10): We felt very sorry for the confusion We have revised it by white-collar (line 261) in the revised manuscript.

11) Overall the discussion could be more succinct but should also acknowledge studies which show contrasting findings to those observed in the current study.

Response 11): We felt grateful for the suggestion and we have revised it in the revised manuscript according to your suggestion.

12) Discussion includes nothing on the proposed mechanisms

Response 12): We would like to thank the reviewer of this suggestion and we have we have elaborated the mechanisms in the Discussion.

13) There are typographical errors in the text, in particular whole words that have been hyphenated e.g., line 27 'dis-ease'

Response 13): We apologized for the editing error and we have revised it in the revised manuscript.

Reviewer #2: 

This manuscript was a very interesting read to me as I have been working on work stress and cortisol research over the last two years. Correlating hair cortisol concentration with hypertension and work stress was a great idea for a scientific manuscript. The statistical analyses appear accurate and sound and the manuscript as a whole is nearly publishable.

The list below includes any concerns I had with the manuscript as it currently stands. Many of them are grammatical in nature. Please be sure to not write in first person (using "we") in scientific literature and don't confuse correlation with causation.

Response: Thanks for the informative comment. We are honored to receive such a high evaluation. The questions you raised about grammar and first person have been revised in the article, and the correlation and causation have been clarified. We will further improve the manuscript and strive to meet higher requirements.

Reviewer Notes

14) Line 27: No reason to have the hyphen in disease.

15) Line 29: Remove the period after “[4]”

Response 14-15): We were very sorry for the mistakes and we have corrected it in the revised manuscript.

16) Line 43: Please be more specific than saying “workers” work in traditionally high-pressure occupations.

Response 16): We have elaborated it specifically in the revised manuscript. (Page 5, Line 47)

17) Line 45: Unsure what you mean by “mental workers”.

Response 17): We felt very sorry for the confusion. We have revised it by white-collar (line 49) in the revised manuscript.

18) Line 52: No need to have a hyphen in researchers

19) Line 70: Change “Compared with in blood…” as you have two prepositions back-to-back. I’d suggest “Compared with cortisol in blood…”

20) Line 88: Don’t write in first person.7

21) Line 90: Add a comma after “study”

Great sample size

22) Line 112: No reason for hyphen in subjects

Response 18-22): We felt very sorry for the error and we have corrected in the revised manuscript.

23) Line 155: Can you better define “irregular” physical exercise?

Response 23): We felt sorry for the confusion and we have clarified in the manuscript. (Page 11, Line 175)

24) Line 156: Can you distinguish “no alcohol consumption” vs “abstinence”?

Response 24): We felt sorry for the confusion. “no alcohol consumption” here means never drinking, while “abstinence” means having a history of drinking but not drinking now.

25) Line 157: Can you describe “smoking cessation”?

Response 25): We felt sorry for the confusion, “smoking cessation”referred to the subjects used to smoke but do not smoke now. We have replaced smoking cessation in the article with “ex-smoker”. (Page 11, Line 178)

26) Line 223 & Line 224: change “level” to “levels”

27) Line 225: Don’t write in 1st person

Response 26-27): We apologized for the editing error and we have modified it in the revised manuscript.

28) Line 232: Move the short paragraph beginning in line 230 into the next paragraph that begins on line 235. Additionally, the two sentences in lines 230-234 are pretty redundant in purpose. I’d suggest editing these two sentences in some way.

Response 28): We have adjusted it in the revised manuscript. (Page 20, Line 255-260)

29) Line 235: What are “mental workers”?

Response 29): “mental workers” means people who rely on their brains to work. Now we have changed the text from mental workers to white-collar (line 265).

30) Line 239: Don’t end a sentence in a helping verb “was”.

31) Line 256: No need for the hyphen in the word service.

32) Line 278: No need for the hyphen in the word under.

33) Line 305: No need for the hyphen in the word required.

34) Line 308: Don’t write in 1st person.

Response 30-34): We were so sorry for these mistakes and we have revised it in the revised manuscript.

35) Line 309 and Line 312: You can’t necessarily say occupational stress caused high HCC (Line 309) or that occupational tension caused high blood pressure (Line 312). You can simply state that they were correlated with one another.

Response 35): We would like to thank the reviewer for the suggestion and we have modified it in the revised manuscript. (Page 24, Line 321-322)

36) Line 312: No need for the hyphen in the word increase.

Response 36): The hyphen in the word have been deleted in the revised manuscript.

---

## [Decision Letter · Decision Letter 1]

12 Apr 2023

PONE-D-22-19724R1A cohort study on the association between changing occupational stress, hair cortisol concentration, and hypertensionPLOS ONE

Dear Dr. Lian,

Thank you for submitting your manuscript to PLOS ONE. After careful consideration, we feel that it has merit but does not yet fully meet PLOS ONE’s publication criteria as it currently stands. Therefore, we invite you to submit a revised version of the manuscript that addresses the points raised during the review process. The reviewers have identified some small open issue which need to be addressed in the revision.Firstly, please address the terminology comment concerning white collar *workers* and please add and mention the difference between alcohol consumption and abstinence in the main body of the manuscript, in the appropriate place.Other than these minor points, I agree with the reviewers that these last updates would render the paper ready to proceed into publication. Please submit your revised manuscript by May 27 2023 11:59PM. If you will need more time than this to complete your revisions, please reply to this message or contact the journal office at plosone@plos.org. Please include the following items when submitting your revised manuscript:A rebuttal letter that responds to each point raised by the academic editor and reviewer(s). You should upload this letter as a separate file labeled 'Response to Reviewers'.A marked-up copy of your manuscript that highlights changes made to the original version. You should upload this as a separate file labeled 'Revised Manuscript with Track Changes'.An unmarked version of your revised paper without tracked changes. You should upload this as a separate file labeled 'Manuscript'.If applicable, we recommend that you deposit your laboratory protocols in protocols.io to enhance the reproducibility of your results. Protocols.io assigns your protocol its own identifier (DOI) so that it can be cited independently in the future. For instructions see: https://journals.plos.org/plosone/s/submission-guidelines#loc-laboratory-protocols. Additionally, PLOS ONE offers an option for publishing peer-reviewed Lab Protocol articles, which describe protocols hosted on protocols.io. Read more information on sharing protocols at https://plos.org/protocols?utm_medium=editorial-email&utm_source=authorletters&utm_campaign=protocols.

We look forward to receiving your revised manuscript.

Kind regards,

Ralph C. A. Rippe, Ph.D.

Academic Editor

PLOS ONE

Journal Requirements:

Reviewers' comments:

Reviewer's Responses to Questions

**Comments to the Author**

1. If the authors have adequately addressed your comments raised in a previous round of review and you feel that this manuscript is now acceptable for publication, you may indicate that here to bypass the “Comments to the Author” section, enter your conflict of interest statement in the “Confidential to Editor” section, and submit your "Accept" recommendation.

Reviewer #1: All comments have been addressed

Reviewer #2: (No Response)

2. Is the manuscript technically sound, and do the data support the conclusions?

Reviewer #1: Yes

Reviewer #2: Yes

3. Has the statistical analysis been performed appropriately and rigorously? 

Reviewer #1: Yes

Reviewer #2: Yes

4. Have the authors made all data underlying the findings in their manuscript fully available?

Reviewer #1: No

Reviewer #2: Yes

5. Is the manuscript presented in an intelligible fashion and written in standard English?

Reviewer #1: Yes

Reviewer #2: Yes

6. Review Comments to the Author

Reviewer #1: The revised paper is easier to follow and the authors have mostly addressed the grammatical issues, one remaining issue is that the authors often refer to 'white collar' rather than 'white collar workers' this needs to be addresed before publication.

Reviewer #2: The only last concern I have is that the "no alcohol consumption" vs "abstinence" still hasn't been addressed within the manuscript itself. As long as this difference can be stated, I think the manuscript should move on towards publication.

7. PLOS authors have the option to publish the peer review history of their article (what does this mean?). If published, this will include your full peer review and any attached files.

Reviewer #1: No

Reviewer #2: **Yes: **Thomas Gerding, MPH

---

## [Author Response · Author response to Decision Letter 1]

21 Apr 2023

Thank you for your very valuable comments and useful suggestions. In response to your suggestions, we do the following point-to-point responses. With your comments and help, we believe that the quality of our articles will be greatly improved. We have reviewed our reference list, revised the format and changes to the reference list have been highlighted in the revised manuscript.

Reviewer #1: 

The revised paper is easier to follow and the authors have mostly addressed the grammatical issues, one remaining issue is that the authors often refer to 'white collar' rather than 'white collar workers' this needs to be addresed before publication.

Response：We felt grateful for the suggestion and we have revised it in the revised manuscript according to your suggestion. (Page 5, Line 51;Page 20, Line 260) 

Reviewer #2: 

The only last concern I have is that the "no alcohol consumption" vs "abstinence" still hasn't been addressed within the manuscript itself. As long as this difference can be stated, I think the manuscript should move on towards publication.

Response: Thanks for the comment. We will further improve the manuscript and strive to meet higher requirements. We have revised it in the revised manuscript. (Page 11, Line 177；Table1)

---

## [Editor Report · Decision Letter 2]

27 Apr 2023

A cohort study on the association between changing occupational stress, hair cortisol concentration, and hypertension

PONE-D-22-19724R2

Dear Dr. Lian,

We’re pleased to inform you that your manuscript has been judged scientifically suitable for publication and will be formally accepted for publication once it meets all outstanding technical requirements.

Kind regards,

Ralph C. A. Rippe, Ph.D.

Academic Editor

PLOS ONE

Additional Editor Comments (optional):

I can see that in the manuscript with Tracked Changes that om p5 and and p20 the addition of the word workers was added, but with strikethrough and marked yellow. In the final submitted version of the manuscript R2 it has disappeared again. Thus, the effort seems to have been made only in the tracked changes version, but it didn't make into the actual final revised manuscript (as in the file list and PDF). Please @authors and @editorial_office check before print! See attached manuscript word doc with notes.

---

## [Editor Report · Acceptance letter]

9 May 2023

PONE-D-22-19724R2 

A cohort study on the association between changing occupational stress, hair cortisol concentration, and hypertension 

Dear Dr. Lian:

I'm pleased to inform you that your manuscript has been deemed suitable for publication in PLOS ONE. Congratulations! Your manuscript is now with our production department. 

Kind regards, 

on behalf of

Dr. Ralph C. A. Rippe 

Academic Editor

PLOS ONE